# Enhanced Thermal Shock Resistance of High-Temperature Organic Adhesive by CF-SiCNWs Binary Phase Structure

**DOI:** 10.3390/ma16175983

**Published:** 2023-08-31

**Authors:** Tingyu Zhao, Zhengxiang Zhong, Xuanfeng Zhang, Jiangfeng Liu, Wenfang Wang, Bing Wang, Li Liu

**Affiliations:** 1MIIT Key Laboratory of Critical Materials Technology for New Energy Conversion and Storage, State Key Laboratory of Urban Water Resource and Environment, School of Chemistry and Chemical Engineering, Harbin Institute of Technology, Harbin 150001, China; 2State Key Laboratory of Advanced Welding and Joining, Harbin Institute of Technology, Harbin 150001, China; 3National Key Laboratory of Science and Technology on Advanced Composites in Special Environments, Harbin Institute of Technology, Harbin 150001, China

**Keywords:** adhesive, thermal shock resistance, ultra-high-temperature ceramics

## Abstract

The development of high-temperature organic adhesive for bonding ultra-high-temperature ceramics with excellent thermal shock resistance has important significance to thermal protection systems for high-temperature environment application. In this study, high-temperature organic adhesive (HTOA) with carbon-fiber-SiC nanowires (CF-SiCNWs) binary phase enhancement structure was prepared. The method is that the SiCNWs grow on the chopped carbon-fiber surface and in the matrix of modified HTOA during high-temperature heat treatment with the help of a catalyst by a tip-growth way and with a vapor–liquid–solid (V-L-S) growth pattern. The results showed that the CF-SiCNWs binary phase enhancement structure plays a significant role in improving thermal shock resistance of high-temperature organic adhesive. The retention rate of the joint bond strength for the bonding samples after 20 cycles of thermal shock testing reaches 39.19%, which is higher than for the ones without CF, whose retain rate is only 6.78%. The shear strength of the samples with the CF-SiCNWs binary phase enhancement structure was about 10% higher than for those without the enhancement structure after 20 cycles of thermal shock.

## 1. Introduction

Ultra-high-temperature ceramics (UHTCs) have great potential for high-performance structural materials in aerospace and aircraft industries due to their remarkable mechanical properties at extremely high temperatures. UHTCs have an extremely high melting point, chemical inertness, abrasion resistance and high strength under high temperature conditions. These excellent properties make them the main structural materials used in high-temperature environments for thermal protection systems, such as rocket engines, aircraft front and other high-temperature parts [1,2,3,4,5]. Unfortunately, due to their intrinsic characteristics and low fracture toughness, it is hard for UHTCs to realize simple and efficient preparation of complex component composites, which limits their further application in the engineering filed [6]. The connection technology is very important for the engineering application of ultra-high-temperature ceramics. Conventional high-temperature structural material joining techniques such as mechanical connection [4], brazing [6,7,8] and inorganic glass connections [9] require complex heat-treatment equipment and tend to cause local degradation of material properties. Bonding ceramic materials with high-temperature resistant adhesive is the most convenient and effective technology [10,11,12,13].

High-temperature-resistant resin is the preferred material for high-temperature binder, which can effectively overcome the problems of high void ratio, mismatch of thermal expansion coefficient, easy stress concentration and low joint strength in the traditional joining technology of ceramic materials [14]. Silicone resin is a polymer having a Si-O-Si bond as a main chain and other organic groups as a side chain. This unique structure allows silicone resin to have both inorganic and organic properties. So, the silicone resin has excellent heat resistance and good low-temperature resistance, and is a potential matrix resin for high-temperature adhesives [12,13]. However, the silicone-resin adhesive system is brittle after heat treatment. Furthermore, under high-temperature conditions, the thermal expansion coefficient of the bonded joint does not match, and large thermal stress is generated at the interface, resulting in poor high-temperature performance and thermal shock resistance of the adhesive [12,15,16]. This is more serious when the temperature reaches 1500 °C. In order to solve the problem, an important toughening and strengthening method is to add fibers, whiskers or nanowires to the adhesive [11,17,18]. One of the main approaches to the mechanical property enhancement of adhesive is to introduce one-dimensional (1D) nanostructures, such as fibers, whiskers and nanowires, due to their great potential for enhancing and toughening adhesive [19,20,21,22,23,24]. The tensile modulus and tensile strength of carbon fiber are about 200 to 700 GPa and about 2 to 7 GPa, respectively; they have been attracting considerable attention due to their remarkable strength, and are widely used as reinforcements for adhesive [20,22]. Otherwise, SiC nanowires with high thermal stability, high specific strength and high toughness are promising reinforcements for adhesive due to their excellent properties [19]. Furthermore, SiC nanowires have important applications in optical devices, nanotechnology and nuclear and space material science [25,26,27]. However, introducing nano-reinforcements brings the challenge of homogeneously dispersing them in the matrix of the adhesive. Since nano-reinforced material has a high specific surface energy, nano-reinforced material is easily agglomerated when introduced [28,29]. Especially if nano-reinforcements are homogeneously distributed in a matrix, structural defects would be generated, thus exerting a bad impact on the mechanical properties of adhesives [30].

Considering the above issue, it is of great importance to propose a new method to introduce nano-reinforcements into the adhesive. Herein, we demonstrate a robust and novel method to enhance thermal shock resistance of high-temperature organic adhesive by CF-SiCNWs binary phase structure. The epoxy emulsion sizing agent is used as the carrier, and the catalyst Ni is supported on the surface of carbon fiber. The SiCNWs grow on the chopped carbon-fiber surface and in the matrix of modified HTOA during high-temperature heat treatment with the help of a catalyst by a tip-growth way and a vapor–liquid–solid growth pattern. The thermal shock-resistance properties and shear strength of enhancement high-temperature organic adhesive by CF-SiCNWs binary phase structure were studied.

## 2. Materials and Methods

### 2.1. Starting Materials

ZrB_2_ powders (99.9%, D = 1 μm) was purchased from Northwest Institute for Non-ferrous Metal Research, Xi’an, China. SiO_2_ and Si powders (99.9%, D = 1 μm) were both purchased from Connaught Qinhuangdao new materials development Co. Ltd., Qinhuangdao, China. B_4_C powder (99.9%, D = 1 μm) was purchased from Aladdin Co. Ltd., Shanghai, China. The Ni particles (99.9%, D = 1 μm) were purchased from Aladdin Co. Ltd., Shanghai, China. The chopped polyacrylonitrile-based carbon fiber (CF) was fabricated in the laboratory, and were 7 μm in diameter and 1 mm (±0.05 mm) in length. Epoxy emulsion sizing agent (HIT-1) and POSS (C_42_H_38_O_12_Si_7,_ 99.9%, M_n_ = 931.34)-MPSR (M_n_ = 3200, solid content = 60%) were fabricated in the laboratory.

### 2.2. Experimental Procedures

In order to significantly improve the adhesion strength and thermal shock resistance of HTOA, the CF-SiCNWs binary phase structure was constructed to synergistically achieve improvement of the two materials. The method is that the SiCNWs grow on the carbon-fiber surface and in the matrix of modified HTOA during high-temperature heat treatment with the help of a catalyst.

#### 2.2.1. Synthesis Modified Carbon Fibers (CFs)

The quantitative Ni powders were added into an epoxy emulsion sizing agent with Ni powders content of 0.5 wt.%, and the mixture was stirred for 30 min to form a homogeneous dispersion. A custom-made small-scale sizing line was employed to size untreated CFs. The CFs were pulled under tension via a nip roller, then passed through a sizing dip bath containing the Ni powders modified sizing agent, and subsequently baked out in the oven.

#### 2.2.2. Synthesis Modified HTOA

The powder mixtures of Ni (as catalysts, 0.5 wt.%), B_4_C (as glass phase) and ZrB_2_ (as high-temperature stabilizers) were subjected to ball-milling with a speed of 400 rpm for 4 h in a polyethylene bottle with ZrO_2_ ball (95%, 0.3–0.4 mm). Then the modified chopped carbon fiber and powder mixtures were added into POSS-MPSR dispersed by ultrasonic for 30 min to prepare modified HTOA.

#### 2.2.3. Preparation of Bonding Samples

Two pieces of ZrB_2_-20 vol% SiC-10 vol% graphite flake (ZSG) composite ceramic with a size of 10 mm × 10 mm × 7 mm were used as bonding samples. The joining surfaces of the samples were polished using SiC paper (<1000 grit) and ultrasonically cleaned in acetone and dried at 80 °C for 30 min. The modified HTOA was spread evenly on one surface of the two ZSGs with a scraper, respectively, and bonded together, and then the high-temperature heat treatment was carried out at 150 °C for 2 h and 1500 °C for 1 h in a stationary argon atmosphere. The heating rate from 150 °C to 1500 °C was 10 °C/min.

### 2.3. Test Methods and Procedure

To explore the thermal stress resistance of the adhesive, thermal shock experiments were performed. After the furnace was heated to 1500 °C by use of an oxyacetylene flame, the samples were put into the furnace and the furnace temperature maintained at 1500 °C for 5 min. Then the samples were taken out of the furnace and kept at room temperature for 1 min. Next, the samples were put into the furnace again for 20 thermal cycles. The isothermal oxidation test of the adhesive samples was carried out at 1500 °C in air in an electrical furnace.

The chemical composition of phase variation of adhesives was analyzed with X-ray diffraction (XRD; Cu Ka radiation D/Max-2500; Rigaku, Akishima, Japan). The chemical element composition of the carbon-fiber surface after loading Ni was measured with X-ray photoelectron spectros copy (XPS, AXIS ULTRA DLD spectromete, KRATOS, Manchester, UK) using a monochromatic Al Kα source (1486.6 eV). High-resolution transmission electron microscopy (TEM, Talos F200x, FEI, Boston, MA, USA) was conducted to characterize the morphologies of self-grown SiCNWs in Modified HTOA. The morphologies of the in-suit construct CF-SiCNWs binary phase structure were analyzed with scanning electron microscopy (SEM, HELIOS NanoLab 600i, FEI Ltd., Boston, MA, USA) with an energy dispersive spectroscopy (EDS). The shear strength was tested under RT by using the Universal Testing Machine (CSS-44001, Changchun, China) at a cross-head speed of 0.2 mm/min depending on GB/T 3357-1982 [31], after the bonding joints were cured at 150 °C and heat treated at 1500 °C for 1 h in argon. The shear strength (τ) was determined using Formula (1):(1)$$\tau =\frac{F_{max}}{A}$$
where $F_{max}$ is the maximum loading load of the sample and *A* is area of bonded surface. Each shear strength value was obtained by taking the average of the three specimens.

## 3. Results and Discussion

### 3.1. Construction and Analysis of Binary Phase Structure

The SEM of the carbon-fiber-loaded Ni powders is shown in Figure 1. The surface of the carbon-fiber precursor is smooth, and the surface roughness of the carbon fiber after the catalyst loading is obviously increased (Figure 1a,b). In addition, there is no obvious agglomeration phenomenon on the surface of the fiber. It can be obtained with XPS analysis (Figure 1c) that the content of nickel on the surface of the fiber is 0.34%, and the sizing agent helps to improve the interfacial adhesion between the carbon fiber and the silicone matrix.

Figure 2a,b are TEM images of self-grown SiCNWs in Modified HTOA. It can be seen from Figure 2a that the SiCNWs obtained from the growth has a diameter of about 100 nm. The corresponding SAED pattern in Figure 2c indicates that the SiCNWs were lattice-striped. The TEM and diffracted spots of high-fold SiCNWs indicate that the lattice spacing is about 0.25 nm in the direction of (111), indicating that SiCNWs were self-growth in HTOA. The typical fracture surface of the adhesive after heat treatment is shown in Figure 2d. It reveals the general morphologies of as-received SiC nanowires. Figure 2e shows the surface microstructure of the adhesive with CF-SiCNWs binary phase structure. It can be seen clearly that the SiC nanowires were randomly grown on the interface of CF with a high density. There was a clear end point in SiCNWs, which was the result of nickel deposition in the SiCNWs’ growth process, indicating that the mechanism of in situ growth of SiCNWs on the surface of carbon fiber was a V-L-S mechanism. As Figure 2e illustrates, CF-SiCNWs binary phase structure was distributed randomly in the pores or on the surface of adhesive. The results of EDS analysis indicate that the elements of SiCNWs grown in situ on the surface of chopped carbon fiber were mainly C element, O element and Si element. The introduction of SiC nanowires could help to afford more stress, which caused the bonding strength of adhesive to increase and the tearing displacement to extend. By interfacial reactions, therefore, CF-SiCNWs binary phase structure was constructed in situ in the adhesive. The specific surface area enlarged with the increase in CF-SiCNWs binary phase structure, causing the enlarging of the joining surface between matrix and whiskers. Additionally, the knots performing like hooks improved the adhesion, which led to the improved mechanical properties of CF-SiCNWs binary phase structure reinforced adhesive.

In this study, SiCNWs grew on the surface of chopped carbon fibers in HTOA by a V-L-S mechanism, thereby introducing SiCNWs on the surface of chopped carbon fibers to form a CF-SiCNWs binary phase structure, improving the heat resistance and seismic performance of HTOA. In order to realize the catalyst participation in the self-growth of SiCNWs on the chopped carbon fiber, the Ni catalyst on the surface of the chopped carbon fiber is designed as shown in Figure 3a. The mechanism of in situ growth of SiCNWs on the chopped carbon fiber in HTOA is shown in Figure 3b. Firstly, Ni powders supported on the surface of carbon fiber in HTOA were gradually melted at a high temperature to form Ni droplets, which became the nucleation center of the next reaction. With the increase in temperature, Si was gradually oxidized into SiO_2_, and then some SiO_2_ could be carbothermal reduced to SiO(g), as described in reactions (2) and (3). The SiC was nucleated as described in reaction (4). The reactions mentioned are given below. Finally, SiO(g) vapors deposited on the SiC and reacted with CO after the nucleation of SiC, and SiCNWs were formed at the CF’s interface initially. Additionally, CO(g) generated by the high-temperature environment of the HTOA matrix and the SiO(g) source is continuously dissolved in the droplet formed in the previous step. When it reaches supersaturation, CO(g) and SiO(g) will gradually precipitate out from the droplet, thereby forming a solid–liquid interface. According to the principle of lowest energy and most stable, SiCNWs generally precipitate from the lowest energy direction, and as this process continues, it will promote the continuous growth of SiCNWs along the one-dimensional direction. Thus, the reaction formulas for growing SiCNWs at 1500 °C are listed below [5,11,19,32]:Si(s) + O_2_(g) = SiO_2_(s),(2)
SiO_2_(s) + C(s) = SiO(g) + CO(g),(3)
SiO(g) + 3CO(g) = SiC(s) + 2CO_2_(g),(4)
CO_2_(g) + C(s) = 2CO(g),(5)

### 3.2. Structure Evolution and Shear Strength of ZSG Joints

The XRD spectra of HTOA samples treated at 1500℃ for 1 h are shown in Figure 4. The adhesive at room temperature was composed of ZrB_2_, C, Si, SiO_2_, Ni and B_4_C. The characteristic peak of SiCNWs appears in the spectrum after the adhesive heat treatment at 1500 °C. The peaks of SiCNWs were observed at 2θ = 35.305°, 41.138° and 60.041°, which was indexed to (111), (200) and (220) reflections of SiC, respectively. At this time, the crystal phase in the HTOA matrix is mainly ZrO_2_, ZrB_2_, ZrSiO_4_, Si, SiO_2_. Since B_4_C is at an ambient temperature greater than 800 °C, it oxidizes to form B_2_O_3_, which is converted to a gaseous state under high-temperature conditions. Therefore, no characteristic peak of B_4_C was observed in the XRD spectrum. In addition, it can be seen from the spectrum that the system has amorphous SiO_2_. This is mainly due to the conversion of the silicone resin into Si-O-C ceramic at high temperature, which is not completely crystallized. Due to the low content of Ni, the peak is smaller. At the same time, the characteristic peaks of ZrO_2_ and ZrSiO_4_ exist in the system. These sharp diffraction peaks indicated high crystallinity. This is because ZrB_2_ reacts with O_2_ to produce ZrO_2_, and ZrO_2_ reacts with SiO(g) in the system to form ZrSiO_4_. At the same time, after heating at 1500 °C for 1 h, there is still a strong ZrB_2_ characteristic peak in the HTOA matrix, indicating that the SiO_2_ glass layer formed by the matrix at a high temperature effectively prevents the diffusion of oxygen into the matrix, and the oxidation of ZrB_2_ is inhibited. Such a strong peak of ZrB_2_ is because of ZrB_2_ in the adhesive as a high-temperature stabilizer. ZrB_2_ has excellent stability under high-temperature conditions, therefore, its content is higher. X-ray diffraction analysis of the sample confirms the conclusion that the SiC nanowires were growing in the adhesive. Thus, the possible reactions at 1500 °C are shown as following [5,11,19,32]:ZrB_2_(s) + 5/2O_2_(g) = ZrO_2_(s) + B_2_O_3_(l),(6)
B_2_O_3_(l) = B_2_O_3_(g),(7)
Si + SiO_2_(l) = 2SiO(g),(8)
ZrO(s) + SiO_2_(l) = ZrSiO_4_(s),(9)
ZrO_2_(s) + SiO(g) + 1/2O_2_(g) = ZrSiO_4_(s),(10)

Figure 5 shows the microstructure of the chopped-carbon-fiber–SiCNWs-composite-toughened HTOA-ZSG joint. It can be seen from Figure 5a,b that after treatment at 1500 °C, the overall structure of the HTOA-ZSG joint is complete and compact. The interface between HTOA and ZSG ceramics is tight and has no obvious defects, indicating that a strong interfacial reaction layer is formed between HTOA and ZSG after a high-temperature interface reaction. At the same time, at 1500 °C, the fillers in HTOA reacted with each other, increasing the compactness of HTOA. There are white and black phases in the HTOA layer, and the white phases may be ZrO_2_ and ZrSiO_4_. In combination with Figure 5b, it can be inferred that the black circular phase in the HTOA is carbon fiber, and the surface of the chopped fiber is embedded by the HTOA matrix, indicating the interface compatibility between the two after high-temperature treatment. Preferably, the stress transfer between the matrix and the chopped carbon fibers can be effectively achieved. By analyzing the element line scan results of the HTOA-ZSG cross-section as shown in Figure 5c, it can be seen that the Ni element can be uniformly distributed in the HTOA and the Zr and O elements have peaks in the EDS curve of the bonding interface. It is possible that the ZrO_2_ and ZrSiO_4_ are formed at a high temperature, and the C element has a peak in some areas of the adhesive layer, where it is possible that the carbon fiber is located. In summary, the chopped carbon fiber has good dispersibility in the matrix. After high-temperature treatment, the SiO_2_ protective layer is formed by the HOTA matrix avoiding the oxidation of the chopped carbon fiber and forming the chopped-carbon-fiber–SiCNWs composite reinforcing and toughening structure.

### 3.3. Fracture Analysis and Toughening Mechanism of ZSG Joints

Figure 6 gives the shear strength of the ZSG joints. It is obvious that the shear strengths of the samples with and without the enhancement phase were 12.8 MPa and 13.2 MPa, respectively. However, the shear strengths of the samples with and without enhancement phase were 37 MPa and 53.8 MPa, respectively, after heating at 1500 °C for 1 h. After treatment of thermal shock, the shear strengths reduce to 3.65 MPa and 14.5 MPa, respectively. The retain rate of the sample with CF reaches 39.19%. Compared with the sample without CF, whose retain rate is only 6.78%, the modified HTOA was significantly improved.

It can be seen from Figure 7 that after the thermal shock test of the HTOA-ZSG joint, the damage of the bonding member mainly occurs in the HTOA rubber layer. The overall structure of HTOA is complete and compact. However, it can be seen from Figure 7b that the HTOA matrix produces holes and cracks because of high-temperature and thermal shock conditions. The generation of the matrix defect becomes a weak point of the joint. When the interface bonding strength is greater than the strength of the substrate, the HTOA matrix itself is destroyed. At the fracture of the HTOA matrix, the carbon-fiber fracture is marked with a red dotted circle and the in situ growth of SiCNWs can be observed at the hole. In summary, the surface-loaded Ni chopped carbon fiber is added to the HTOA, and the CF-SiCNWs binary phase structure is constructed in situ in the system. In situ growth of SiCNWs in the pores of the HTOA can effectively improve the thermal shock resistance of HTOA.

The reasons for CF-SiCNWs binary phase structure improving the thermal shock resistance of HTOA are as follows: the CF-SiCNWs binary phase structure enhances the high-temperature resistant adhesive, reduces the thermal expansion coefficient of the system, achieves the thermal expansion matching of the HTOA with the bonded substrate and reduces the thermal stress generated during thermal shock. In addition, when stress is transferred in adhesive matrix, the nanowires could easily become crack origins because of the high stress accumulation. Through in situ synthesizing of nanowires with high strength and specific surface, fracture energy could be enlarged. During crack propagation, the distinct abrupt deflection will be shown by cracks near the nanowires, due to residual stress expanding near the nanowires. If the drive cannot support the crack propagation, the nanowires will pin the cracks. However, with further enhancement of the external load, sufficient energy can be provided, and the cracks will be turned to another direction.

## 4. Conclusions

In summary, we have demonstrated a robust and novel method to enhanced thermal shock resistance of high-temperature organic adhesive by CF-SiCNWs binary phase structure. The chopped carbon fiber and SiCNWs were uniformly introduced into the HTOA, and the SiCNWs were synthesized in situ on the carbon-fiber surface and the pores of HTOA using carbon and silicon sources generated from the heat treatment of the HTOA during heating at 1500 °C. The length of SiCNWs on the surface of chopped carbon fiber increases with the increase in temperature. The chopped-carbon-fiber–SiCNWs composite structure improves the thermal shock resistance of HTOA. The joint bond strength after heat treatment at 1500 °C/1 h is 37 MPa. After thermal shock testing, the bond strength is 14.5 MPa, and the strength retention rate reaches 39.19%. Otherwise, the shear strength of the samples with the CF-SiCNWs binary phase enhancement structure was about 10% higher than for those without the enhancement structure after 20 cycles of thermal shock.

## Figures and Tables

**Figure 1 materials-16-05983-f001:**
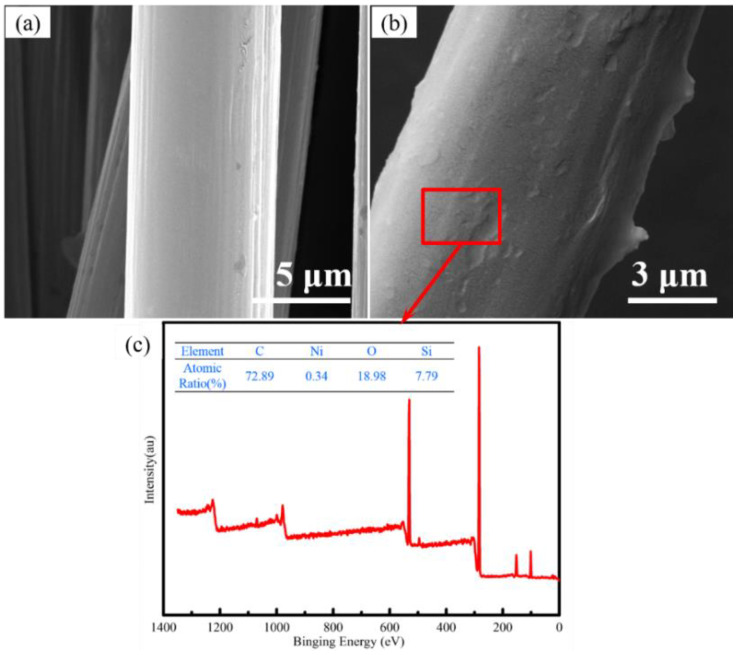
SEM images of the chopped carbon-fiber surface before and after loading Ni. (**a**) protofilament; (**b**) loading Ni; (**c**) XPS.

**Figure 2 materials-16-05983-f002:**
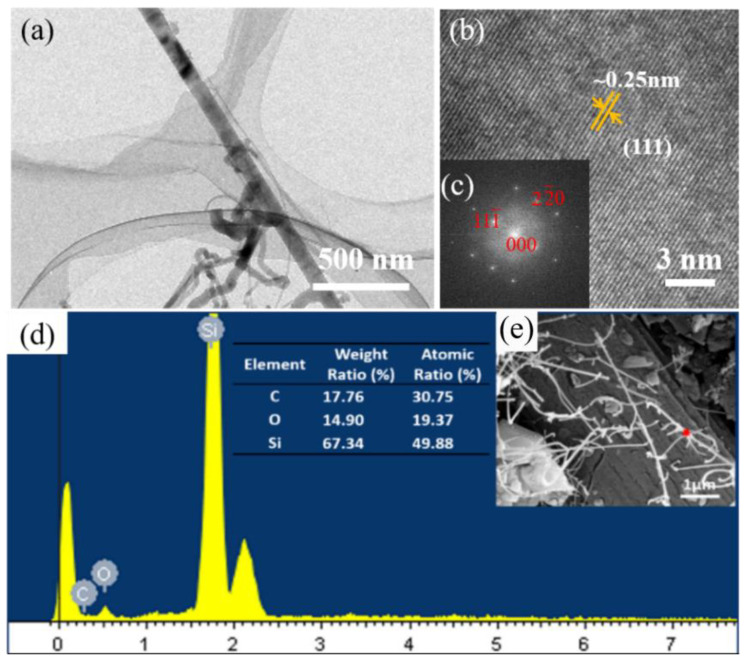
Micrographs and structure analysis of modified HTOA with in situ construct CF-SiCNWs binary phase structure: (**a**–**c**) low-magnification TEM image and the corresponding SAED pattern of the synthesized SiCNWs; (**d**,**e**) SEM micrographs and EDS analysis at the red dot.

**Figure 3 materials-16-05983-f003:**
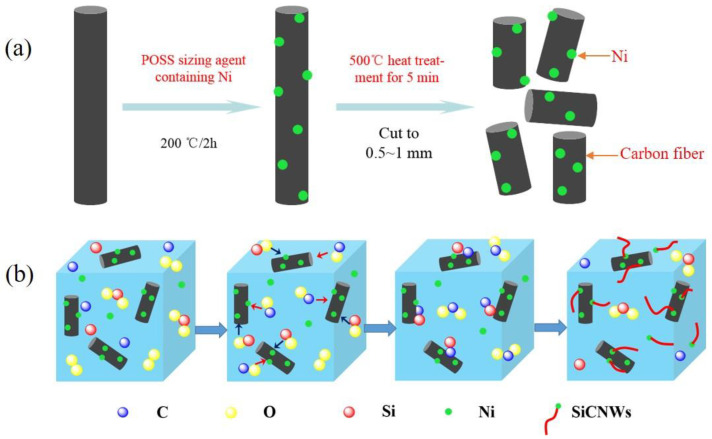
(**a**) Schematic diagram of catalyst for the loading Ni on chopped carbon-fiber surface reaction; (**b**) Schematic diagram of in situ construct CF-SiCNWs binary phase structure in HTOA.

**Figure 4 materials-16-05983-f004:**
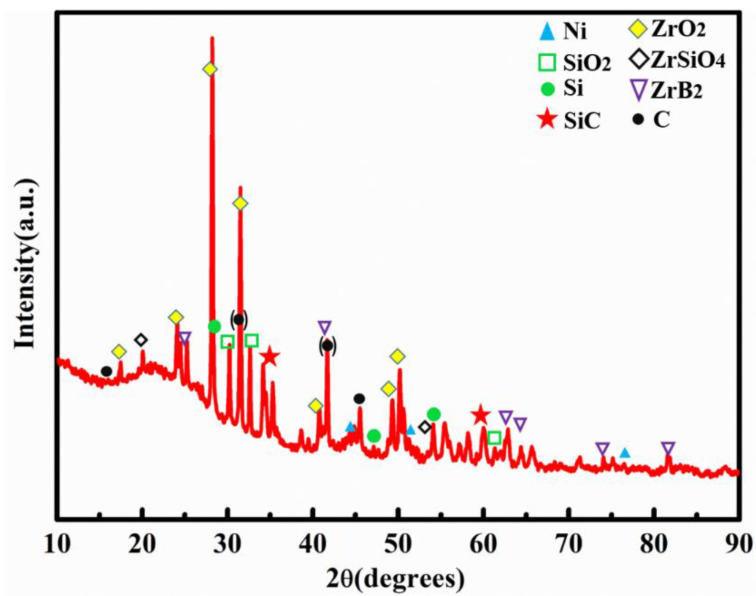
XRD patterns of the HTOA after heat treatment at 1500 °C for 1 h in air.

**Figure 5 materials-16-05983-f005:**
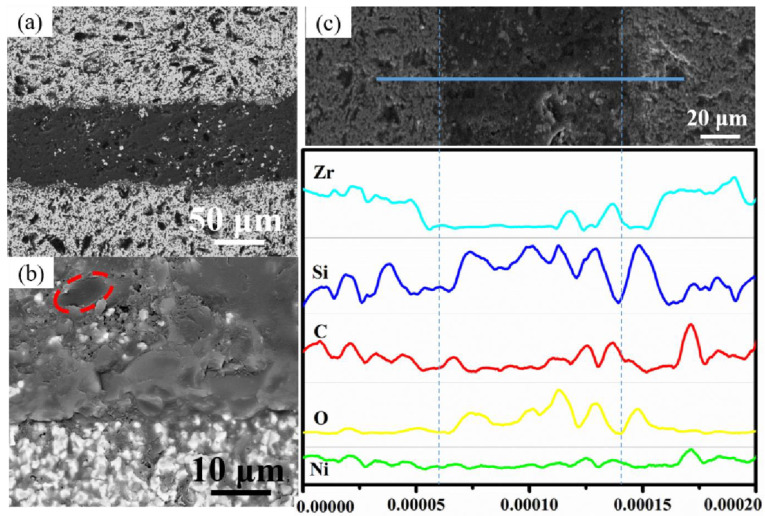
The microstructure and element distribution in interlayer ZSG ceramic joints. (**a**) microstructure of ZSG joint after heat treatment at 1500 °C for 1 h in air; (**b**) microstructure of carbon fiber (red dotted line) compatible with HOTA interface; (**c**) the EDS line scanning result of the joint after heat treatment.

**Figure 6 materials-16-05983-f006:**
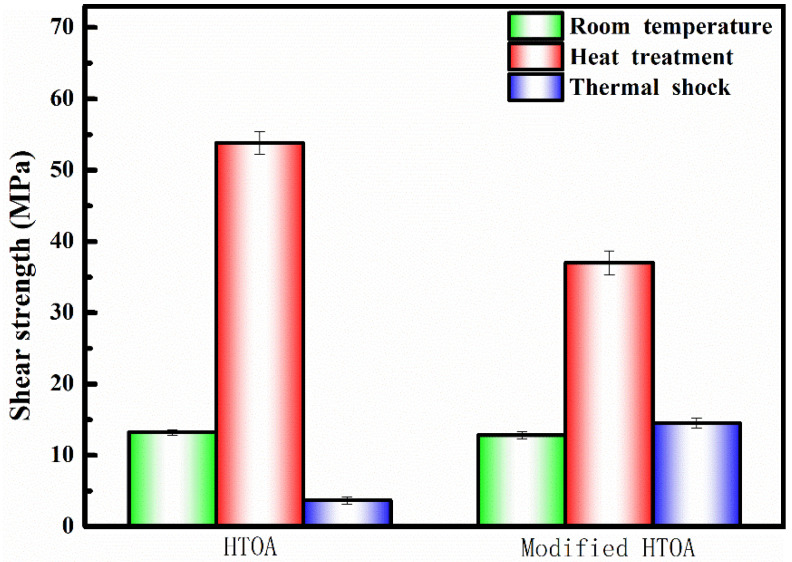
Shear strength of ZSG joints under different processing conditions.

**Figure 7 materials-16-05983-f007:**
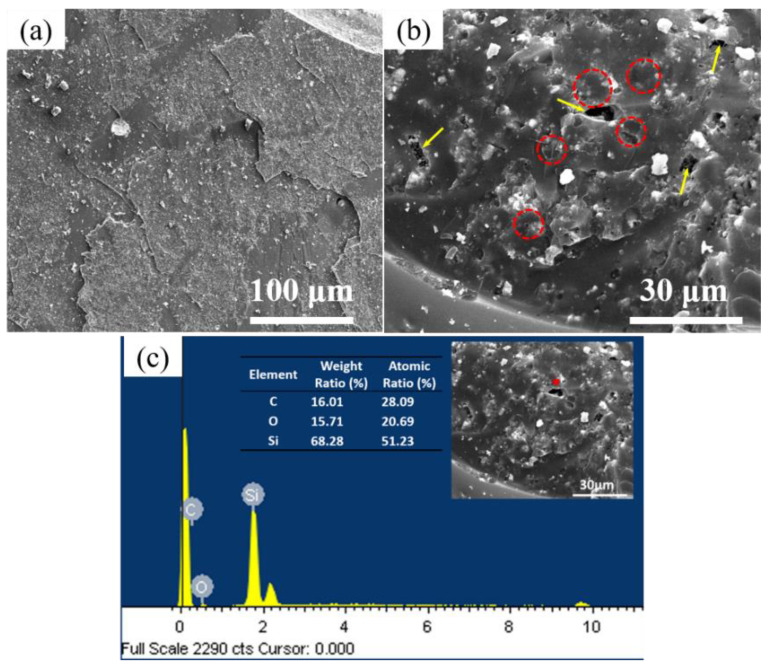
SEM and EDS analysis of the fracture of the sample after the shear strength test. (**a**) SEM image of the HTOA rubber layer; (**b**) holes (yellow arrows) and cracks (red dashed circles) of the HTOA matrix; (**c**) EDS analysis at the red dot.

## Data Availability

Not applicable.

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
