# Peer review of "Enhanced Thermal Shock Resistance of High-Temperature Organic Adhesive by CF-SiCNWs Binary Phase Structure"

_materials, 2023, doi:10.3390/ma16175983_

Round 1
Reviewer 1 Report
Referee report on “Enhanced Thermal Shock Resistance of High Temperature Organic Adhesive by CF-SiCNWs Binary Phase Structure”
Although this topic is of some interest, this manuscript in its present form cannot be recommended for publication and requires some improvement and clarification.
1. From the Introduction it is not clear what crystalline modification of silicon carbide is meant.
2. Furthermore, the introduction needs more general information about SiC and SiC nanowires and its important applications in optical devices, nanotechnology and nuclear and space material science. This is important to attract more reader interest and further incentive applications. For some of them, see, for example:
a) Huczko, A., Dąbrowska, A., et al. Silicon carbide nanowires: synthesis and cathodoluminescence. physica status solidi (b), 2009, 246(11‐12), 2806-2808. https://doi.org/10.1002/pssb.200982321
b) Lebedev, A.S., et al. Carbothermal Synthesis, Properties, and Structure of Ultrafine SiC Fibers. Inorg Mater 56, 20–27 (2020). https://doi.org/10.1134/S0020168520010094
c) Tynyshbayeva, K.M.; et al. Study of Helium Swelling and Embrittlement Mechanisms in SiC Ceramics. Crystals 2022, 12, 239. https://doi.org/10.3390/cryst12020239
3. Line 142. “nickel on the surface of the fiber is 0.34%” - this accuracy requires justification and measurement errors.
4. Fig. 4. It would be extremely interesting to see both Raman and luminescent measurements simultaneously as support.
In general, the manuscript is interesting and can be considered for publication after constructive reflection on the above comments.
Reviewer 2 Report
1- The correct form to write the unit of measurement is "min" instead "minutes".
2- XPS and TEM characterization are not indicated in section 2.3 and must be included
3- The article is very well explained from the chemical point of view, and the analysis of the properties has also been carried out in a very complete way. They should only correct minor details in the document
Reviewer 3 Report
It is an interesting work , but some clarifications and corrections are needed

Reviewer 4 Report
The manuscript entitled “Enhanced Thermal Shock Resistance of High Temperature Organic Adhesive by CF-SiCNWs Binary Phase Structure” describes how one can improve the thermal shock resistance of high-temperature organic adhesive (with carbon-SiC nanowires). The manuscript can be of high interest to a potential reader. It is well-prepared and written. In the present version, a potential reader can find some misprints, lack of information, etc., but these errors do not detract from the manuscript’s importance. Thus, I recommend publishing the manuscript in the Journal after minor amendments.
1. Page 2, lines 84-91: Add information about the purities of the powders bought.
2. Page 2, line 90: Are the values of the diameter and length of fabricated carbon fiber the mean ones? If the answer is positive, the authors can add some information about the standard deviations from the mean values.
3. Figures 1, 2, 4, 5, 6, 7: Add information about the temperature of the measurements.
4. Sections 2.2.1; 2.2.2, XRD and SEM measurements (section 2.3): Add information about the temperature. There is no information about the balls used in the milling procedure.
5. Page 4, line 155; and Figure 2: There is no figure 2e. Is it an inserted figure to figure 2d? There is no caption for figure 2e.
6. The caption of figure 7: Add information about the red dashed coil and the yellow arrows used there.
7. The minor errors and misprints found: (i) page 1, line 24: The sentence starts from “And” – it can be omitted, and the sentence should be “The shear strength (…)”; (ii) page 4, line 159: Give the full name after the first use of the acronym “V-L-S” (vapor-liquid-solid; by the way, the full name is given in the abstract section – page 1, line 20 – but there is no acronym given); (iii) page 5, line 179: the authors used the acronym VLS, compare with V-L-S (the same one) used on page 4, line 159); (iv) page 6, lines 188-189; add, for clarity and to inform a potential reader, that the reactions mentioned there are given below; (v) page 6, line 205: the “degree symbols” are missing there; (vi) figure 5c: the authors can point out the positions of the interface on a figure presented there; (vi) page 8, line 253; the degree symbol should be as the superscript.
Round 2
Reviewer 1 Report
The authors have sufficiently improved their original manuscript, which now can be recommended for publication.
